# Design of Reflective Polarization Rotator in Silicon Waveguide

**DOI:** 10.3390/nano12203694

**Published:** 2022-10-21

**Authors:** Li-Ying Liu, Hong-Chang Huang, Chu-Wen Chen, Fu-Li Hsiao, Yu-Chieh Cheng, Chii-Chang Chen

**Affiliations:** 1Department of Optics and Photonics, National Central University, Taoyuan 320317, Taiwan; 2Institute of Photonics, National Changhua University of Education, Changhua 500207, Taiwan; 3Department of Electro-Optical Engineering, National Taipei University of Technology, No. 1, Sec. 3, Chung-Hsiao E. Rd., Taipei 106344, Taiwan

**Keywords:** polarization rotator, birefringent, photonic crystals, waveguide

## Abstract

In this work, we investigate theoretically the reflective polarization rotator in a silicon waveguide formed by periodically arranged rectangular air holes. The etched air holes generate the large birefringence for the waveguide. The effective refractive index of the non-etched waveguide is isotropic. The structure can be regarded as a stack of alternating birefringent waveplates and isotropic material similar to the folded Šolc filter. The band structure of the stack of birefringent waveplates with isotropic background is calculated to confirm the fact that high reflection peaks in the reflection spectra of the waveguide result from the photonic bandgap. The polarization extinction ratio for the reflected light is 15.8 dB. The highest reflectivity of the device is 93.1%, and the device length is 9.21 μm. An ultra-wide operation bandwidth from 1450.3 to 1621.8 nm can be achieved.

## 1. Introduction

Polarization rotators [1,2,3,4,5] are an essential component in optical fiber communication systems for applications such as the circulator [6,7]. The polarization rotators formed by metasurfaces or metamaterials have also been studied [8,9,10,11,12,13]. Free-space metasurface polarization rotators have also been developed for microwaves [14,15]. The phase of the rotated waveform can be shifted with the help of the PIN diode, which provides the ability to modify the scattering properties of the metasurface [8]. For the integrated optical polarization rotators, asymmetric cross-sections of the waveguide produce the horizontal and vertical polarization modes. The birefringence of the two eigenmodes of the waveguide, which are oriented at 45° with respect to the wafer axis, results in the coupling between two eigenmodes and the polarization rotation [1,2,3,4,5]. In high-power, short-pulse laser systems, high-precision reflective polarization rotators are required to provide a higher accuracy and a higher damage threshold [16]. A reflective polarization rotator in a silicon waveguide with a non-vertical waveguide sidewall and anti-symmetric grating structure has been developed [17]. The polarization extinction ratio of the device at 15 dB could be attained for the grating length of 50 μm. The corrugation period was 354 nm. The corresponding period number was 141. The 3 dB bandwidth in the reflection spectra was around 18 nm at the center wavelength of 1460 nm. The maximum reflectivity is around −2.5 dB (56%). We recently reported two novel transmission-type polarization rotators based on shifted circular and rectangular air holes in a silicon waveguide [18,19], which could be used in quantum computing as the logic gates. The device length can be shorter than the L-shaped or trench waveguides [20,21,22,23] due to the higher birefringent effect formed by photonic crystals. In the present work, we propose a reflective polarization rotator in a silicon waveguide formed by periodically arranged L-shaped silicon waveguides. The etched air holes induce the large birefringence for the waveguide. The non-etched waveguide can be regarded as an isotropic material. The structure is similar to a folded Šolc filter with an anisotropic periodic dielectric stack [24]. In the literature, Abdulhalim reported a reflective polarization rotator for the first time by using a folded Šolc structure with a 45-degree twist and high birefringence [25,26]. In Ref. [25], the reflective polarization rotator of Abdulhalim was demonstrated near the photonic bandgap where the omnidirectional reflection [27,28,29,30] occurs. Additionally, in Ref. [26], it was demonstrated as resonant reflective peaks of a Fabry–Pérot structure. While the reflective polarization rotator of Abdulhalim was the first reported using a folded Šolc structure, ours is the first for an integrated optical waveguide form utilizing properties of photonic crystal structures. The layer thicknesses, high birefringence, and 45-degree twist were the conditions postulated by Abdulhalim to observe the reflective polarization rotator. Here, our photonic crystal structure follows Abdulhalim’s design with the twist angle determined by the waveguide mode. In the present study, by calculating the band structure, the high reflection wavelength range in the reflection spectra of the waveguide is confirmed to originate from the photonic bandgap of the structure. The 3 dB bandwidth of the high reflection wavelength range can be as large as 171.5 nm at the central wavelength of 1550 nm, which covers the wavelength range from 1450.3 to 1621.8 nm.

## 2. Reflective Polarization Rotator in a Silicon Waveguide

The reflective polarization rotator is formed by etched rectangular air holes periodically arranged in a silicon waveguide on SiO_2_ substrate. The schematic drawing of the device is illustrated in Figure 1. The cross-section of the L-shaped waveguide formed by the etched rectangular air hole is also illustrated. The refractive indices of the silicon and SiO_2_ are 3.46 and 1.46, respectively. Both the width and height of the silicon waveguide are 350 nm to provide two waveguide modes (the fundamental mode and the first-order mode). Both the depth and the width of the etched rectangular air holes are defined to be half of the waveguide height and width, i.e., 175 nm. The etched rectangular air holes forming the L-shaped waveguide provide the birefringent effect [20]. The duty cycle (DC = the length of the etched waveguide/period = d/Λ) is 50%. The corrugation rectangular air holes form the stack of birefringent waveplates with isotropic background similar to the folded Šolc filter structure. The slow and fast axes of the birefringent waveplates are illustrated schematically in Figure 1.

The mode profile of the non-etched waveguide at wavelength 1550 nm (λ) in the TM mode (E_Y_) is calculated by the beam propagation method [31]. The polarization direction of the electric field of the TE and TM modes is parallel and perpendicular to the sample top surface, respectively. The mode profile in the TM mode (E_Y_) is launched into the reflective polarization rotator. The propagation of light in the silicon waveguide together with the light transmission and reflection from the corrugation rectangular air holes are analyzed for the TE (E_X_) and the TM (E_Y_) modes using the three-dimensional eigenmode expansion method [32]. The period Λ is scanned from 0.1 to 1.2 μm. For the period between 0.1 and 0.5 μm, the scanning step and the spatial grid size are 0.002 and 0.002 μm, respectively. For the period between 0.5 and 1.2 μm, the scanning step and the spatial grid size are 0.0035 and 0.007538 μm, respectively. The corresponding normalized frequency Λ/λ is from 0.0645 to 0.7742. The duty cycle of the corrugation rectangular air holes is varied from 0.1 to 0.9. The number of periods is 30. The normalized reflection spectra of the TE mode (R_X_) are shown in Figure 2a. Figure 2b shows the reflection spectra for the peaks at the normalized frequency around 0.12. We can observe that for DC = 0.5 and Λ/λ = 0.119, the reflection for the TE mode (R_X_) is as high as 85.29%. The transmission for the total energy of both the TE and TM modes is also characterized to be 8.70%. The reflection for the TM mode (R_Y_) is 2.3%. The corresponding round-trip propagation loss is 3.71%. The polarization extinction ratio, which is defined as
(1)$$10\times \log_{10}\left(\frac{R_{X}}{R_{X}+R_{Y}}\right)$$
is −15.8 dB. The total length of the etched zone, consisting of 30 periods of air holes, is 5.53 μm. If the wavelength of the peak maximum is 1550 nm for Λ/λ = 0.119 and DC = 0.5, then the corresponding period Λ is 184.45 nm. The wavelength range at half maximum is from 1450.3 to 1621.8 nm, showing an ultra-wide operation bandwidth of 171.5 nm for the reflective polarization rotator in the silicon waveguide.

In Figure 2a, for Λ/λ around 0.34 and 0.57, we can also observe the reflection peaks. In our structure, the etched air holes induce the large birefringence for the waveguide. The effective refractive index of the non-etched waveguide is isotropic for different polarizations in the propagation direction. The structure can be regarded as a stack of alternating birefringent waveplates and the isotropic material, which is similar to the folded Šolc filter. To understand the origin of the peaks in the reflection spectra of the birefringent photonic crystal structure formed by the corrugation rectangular air holes in the silicon waveguide, we carried out the 4 × 4 matrix method [24,33] for the stack of birefringent waveplates with isotropic background.

## 3. Polarization Rotator Formed by the Stack of Birefringent Waveplates with Isotropic Background

In our design, the etched rectangular air holes in the silicon waveguide is also called the L-shaped waveguide, in which the large birefringent property has been found [23], as illustrated in Figure 1. The corrugation rectangular air holes and non-etched waveguide can be regarded as alternating birefringent waveplates and isotropic material, respectively. The structure is similar to the folded Šolc filter, which consists of only a stack of alternating birefringent waveplates. The equivalent schematic drawing of a stack of alternating birefringent waveplates and surrounding isotropic material is illustrated in Figure 3. The effective indices of the silicon waveguide with etched rectangular air hole for the first two modes are calculated by the beam propagation method to be 2.1662 and 1.8971, respectively. These two values are used for the refractive indexes of the slow and fast axes of the birefringent waveplates, respectively. The effective index of the non-etched silicon waveguide is 2.2950. We use this value for the isotropic material between the birefringent waveplates. Since the difference between the effective refractive index of the waveguides with the etched holes and that without etched air holes is in the order of magnitude of 10^−1^ [20]; the reflection between the plates should be considered in this work. The E_Y_-polarized light is launched into the structure, as shown in Figure 3. The angle between the fast axis of the birefringent waveplates and the X-axis is θ. The 4 × 4 matrix method [24] is adopted to calculate the reflection spectra of the stack of birefringent waveplates with isotropic background. The structure of birefringent waveplates with different duty cycles from 0.1 to 1 is analyzed. The period number is 30. In the case of the silicon waveguide with 30 etched holes (duty cycle = 0.5), the reflectivity ratio of the E_X_-polarized to E_Y_-polarized light (R_X_/R_Y_) is 17.59 at the normalized frequency of Λ/λ = 0.119. In the case of the birefringent waveplates, by scanning the angle θ between the fast axis and the X-axis, the reflectivity ratio of the E_X_-polarized to E_Y_-polarized light (R_X_/R_Y_) is shown in Figure 4. As the angle θ between the fast axis and the X-axis is 26.9°, the reflectivity ratio of the E_X_-polarized to E_Y_-polarized light (R_X_/R_Y_) is identical to that calculated by the silicon waveguide (17.59). With the optic axis θ at 26.9°, the reflection spectra of the E_X_ for the stack of birefringent waveplates with isotropic background are calculated as shown in Figure 5. We can observe that at Λ/λ = 0.12, 0.34, and 0.57, the reflection peaks appearing in Figure 2a can also be found in Figure 5.

Using the 4 × 4 matrix method, the band structure of the stack of birefringent waveplates with isotropic background is calculated by solving the eigenvalue of the Bloch waves [24,33,34]. The precise band edge can be obtained. The duty cycle is 0.5. Figure 6a illustrates the band structure. The red and the blue curves present the dispersion of the slow and fast waves, respectively. The band edges of the first bandgap are at the normalized frequencies of 0.1127 and 0.1177, respectively. At these frequencies, the slow and fast waves meet the exchange Bragg condition [24], K_s_Λ + K_f_Λ = 2 mπ, where K_s_ and K_f_ are the Bloch wave numbers for the slow and fast waves, respectively; m is an integer. The coupling (exchange) of the slow and fast waves or the so-called rotation of the polarization occurs at these band edges. The band edges of the second bandgap are at the normalized frequencies of 0.2210 and 0.2337 for the slow axis. The band edges of the second bandgap are at the normalized frequencies of 0.2232 and 0.2466 for the fast axis. At these frequencies, the polarization of the reflected light is identical to that of the incident light since KΛ is null where K is the Bloch wave number. The band edges of the third bandgap are at the normalized frequencies of 0.3430 and 0.3509, respectively. The band edges of the fifth bandgap are at the normalized frequencies of 0.5771 and 0.5781, respectively. The first, third, and fifth bandgaps correspond to the frequency ranges of the peaks in the reflection spectra shown in Figure 2a and Figure 5. This indicates the fact that the reflection of the polarization rotator in the silicon waveguide originates from the photonic bandgap, and that the wavelength range of the reflection peaks can be estimated by calculating the band structure of the stack of birefringent waveplates with isotropic background.

## 4. Discussion

The band edges of the first bandgap are shown in Figure 6b for different duty circles. The red and blue solid curves present the upper and lower band edges of the first band, respectively, for different duty cycles. We can observe that the bandwidth increases with increasing duty cycle. For the duty cycle of 100%, the bandwidth is 0.0089. The red and blue circles present the frequencies at the half maximum for the peak at around Λ/λ = 0.11~0.13, as shown in Figure 2b, which are calculated from the silicon waveguide with etched holes. We can observe that the bandwidth is less sensible to the duty cycle. For the duty cycle of 50% and 100%, the bandwidth is 0.0134 and 0.012, respectively. All of the bandwidths for different duty cycles obtained from the silicon waveguide (red and blue circles) are roughly equal to those for the duty cycle of 100%, which was acquired from the band structure (0.0089). This may originate from the fact that the conversion from the asymmetrical mode of the etched waveguide into the symmetrical mode of the non-etched waveguide requires a sufficient length for light propagation. This effect results in an asymmetrical mode profile in the non-etched waveguide, inducing the birefringent effect in the non-etched waveguide. This phenomenon is different from the stack of birefringent waveplates with isotropic background in which the wave keeps the polarization state unchanged in the isotropic material. Figure 7 illustrates the reflection of the E_X_-polarized light in the etched silicon waveguide with a duty cycle of 50% for different numbers of periods from 20 to 50. The reflection increases with the number of periods. With 50 periods, the reflectivity can be as high as 93.1%, indicating that good polarization conversion efficiency can be achieved by the reflective polarization rotator in the silicon waveguide. The corresponding device length is 9.21 μm, which is much shorter than that reported in Ref. [17] (50 μm).

The dimension (length, width, and depth) error of the etched rectangular trench may induce the discrepancy between the device performance and the simulation results. The length error could result in a change in the duty cycle, leading to a change in the transmission, frequency, and bandwidth of the peak, which can be observed in Figure 2b and Figure 6b. The error of the width and depth of the etched trench could change the eigenmode pattern and the effective index of the etched waveguide, leading to the change in the peak frequency.

Corner rounding can also often occur for the rectangular trench during the etching process during device fabrication. At the normalized frequency Λ/λ of 0.12, the wavelength (λ) of the peaks is mainly influenced by the period of the pattern (Λ). The feature size of corner rounding is relatively too small in comparison with the wavelength to influence light propagation. For the higher normalized frequency, where the wavelength is shorter, the feature size of corner rounding becomes significant to the wavelength. Corner rounding can influence the normalized frequency of the peaks. In other words, the fact that corner rounding changes the higher spatial frequency components of the rectangular trench pattern leads to influencing the frequency of the peaks at higher normalized frequency range.

## 5. Conclusions

In this work, we propose the reflective polarization rotator in a silicon waveguide formed by corrugation rectangular air holes. When the period number is 50, the reflection of the etched silicon waveguide for the E_X_-polarized light can be as high as 93.1%. The corresponding device length is 9.21 μm. The ultra-wide operation bandwidth can cover from 1450.3 to 1621.8 nm. The bandgap of the periodically arranged etched air holes forming the stack of birefringent waveplates with isotropic background is studied using the 4 × 4 matrix method. We show that the reflection spectra and the photonic bandgap can be obtained. This method can facilitate the device design for further applications. The polarization rotator can served as a Pauli–Z gate in quantum computing.

## Figures and Tables

**Figure 1 nanomaterials-12-03694-f001:**
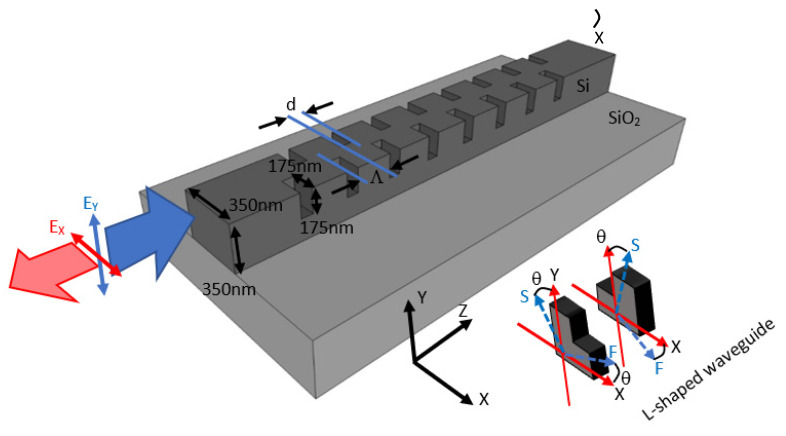
Reflective polarization rotator in the silicon waveguide with corrugation rectangular air holes. The cross-section of the L-shaped waveguide formed by the etched rectangular air hole is presented. The *S*-axis and *F*-axis are the slow and fast axes, respectively.

**Figure 2 nanomaterials-12-03694-f002:**
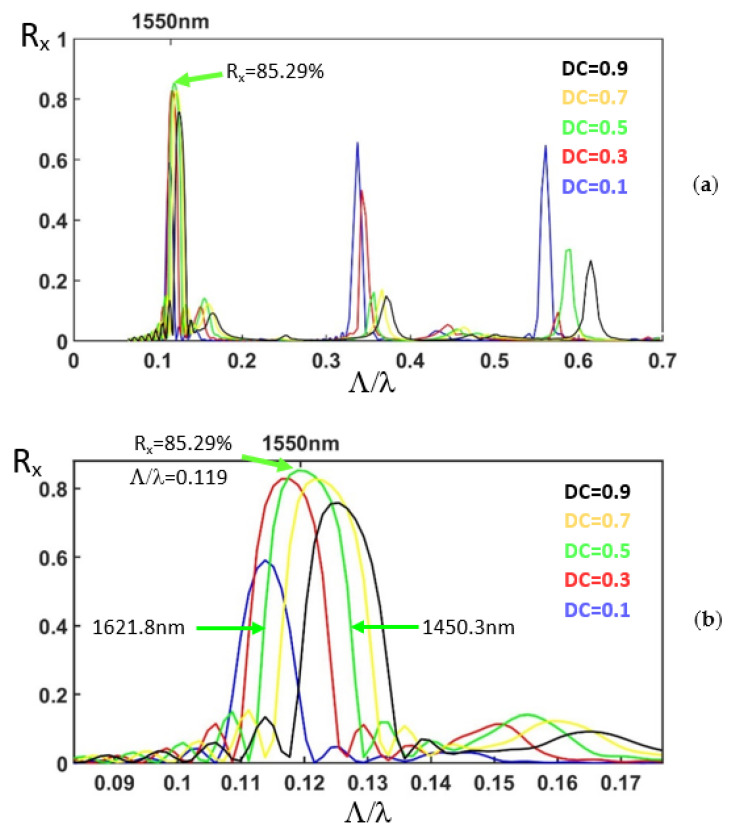
Reflection spectra for the TE mode (E_X_) as the light of the TM mode (E_Y_) is launched. The number of periods is 30. The horizontal axis is the normalized frequency (**a**) from 0 to 0.7 (**b**) from 0.08 to 0.175.

**Figure 3 nanomaterials-12-03694-f003:**
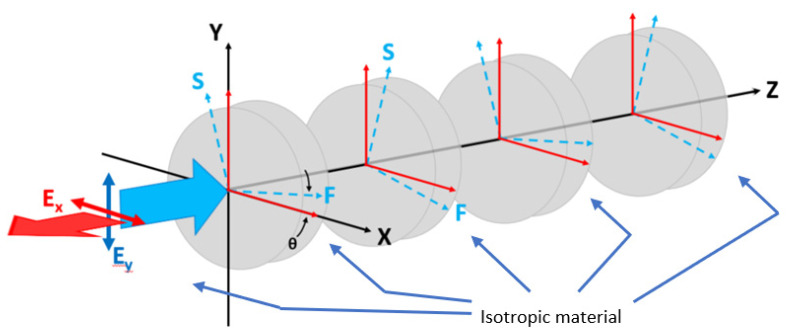
A stack of alternating birefringent waveplates. The material between the waveplates is isotropic.

**Figure 4 nanomaterials-12-03694-f004:**
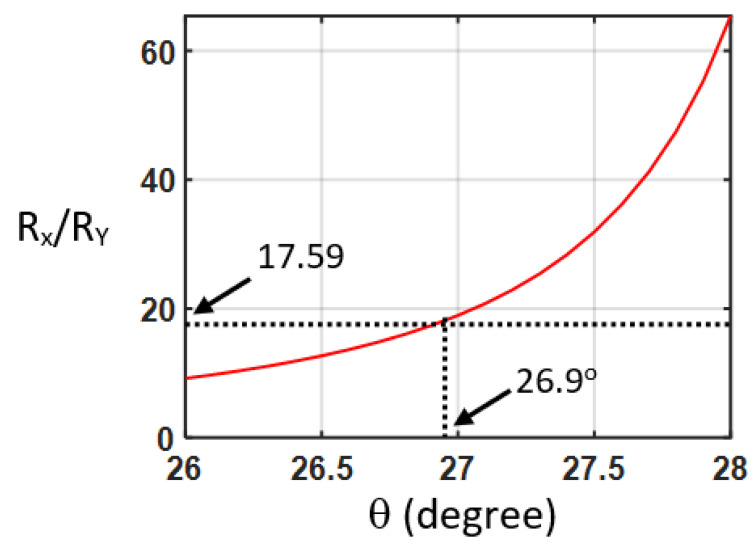
Reflectivity ratio of the E_X_-polarized to E_Y_-polarized light for the stack of birefringent waveplates with isotropic background for different θ angles between the fast axis and the X-axis at the normalized frequency of Λ/λ = 0.119.

**Figure 5 nanomaterials-12-03694-f005:**
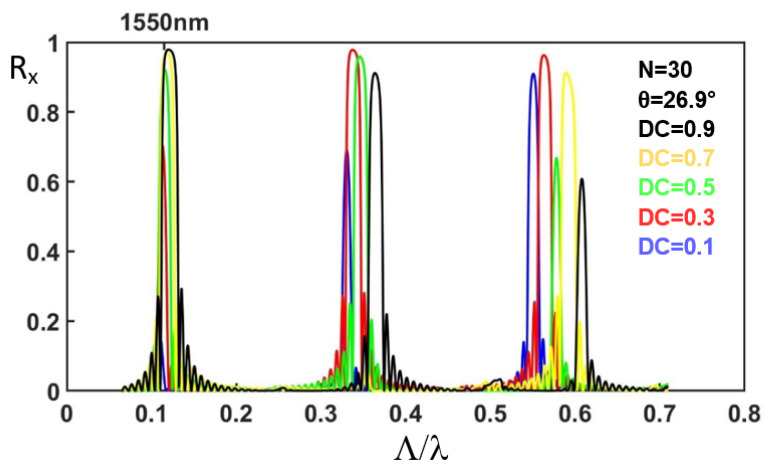
The reflection spectra of E_X_ of the stack of birefringent waveplates with isotropic background for different duty cycles. The incident light is linearly polarized in the Y-direction. The period number is 30. The angle between the fast axis and the X-axis is 26.9°.

**Figure 6 nanomaterials-12-03694-f006:**
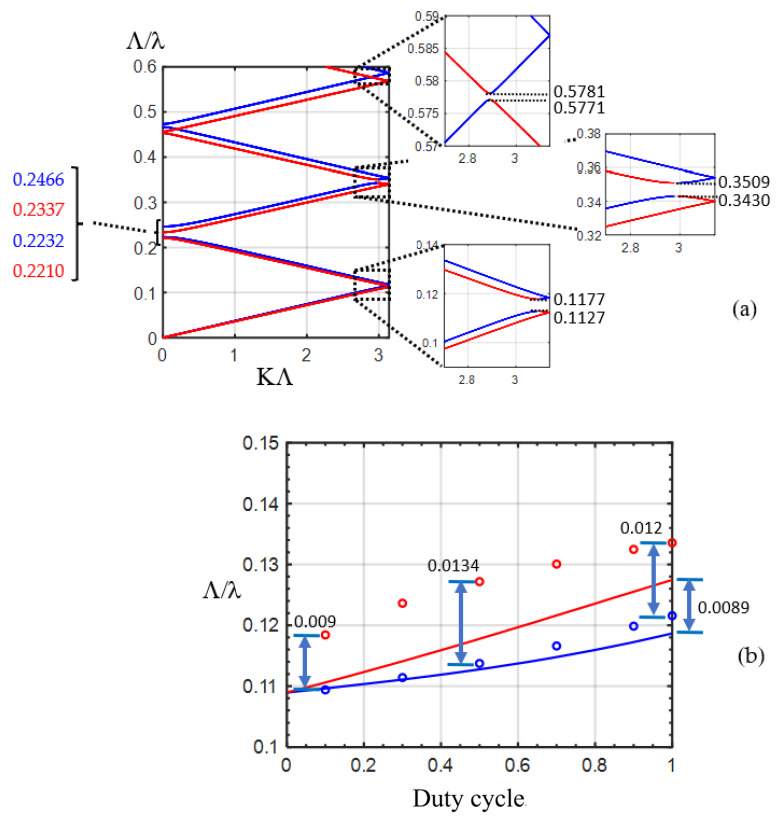
(**a**) Band structure of the stack of birefringent waveplates with isotropic background for a duty cycle of 0.5. (**b**) The band edges of the first bandgap for different duty cycles. The solid lines are obtained from the band structure. The circles are acquired from the frequencies at half maximum of the peak at around Λ/λ = 0.12, as shown in Figure 2b.

**Figure 7 nanomaterials-12-03694-f007:**
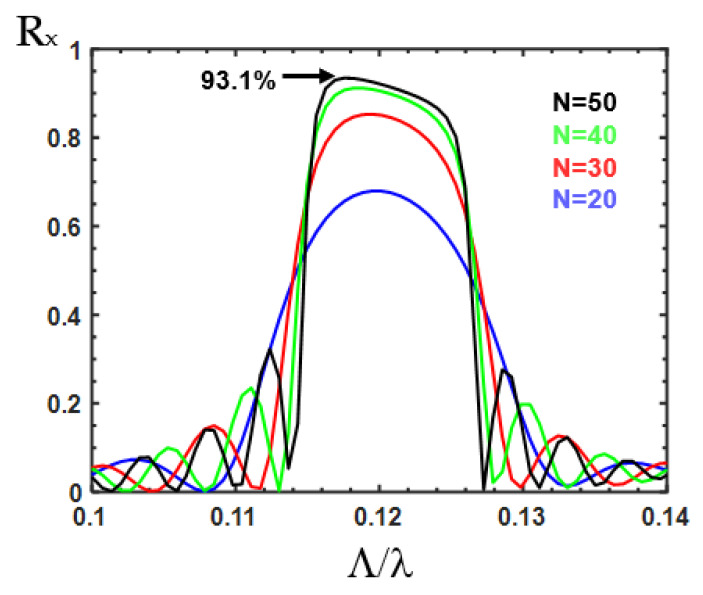
Reflection spectra of the E_X_-polarized light in the reflective polarization rotator in the silicon waveguide for different numbers of periods.

## Data Availability

Not applicable.

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
