# Peer review of "Design of Reflective Polarization Rotator in Silicon Waveguide"

_nanomaterials, 2022, doi:10.3390/nano12203694_

Round 1
Reviewer 1 Report
In the manuscript entitled “Reflective Polarization Rotator in Silicon Waveguide”, L.-Y. Liu et al. have proposed a reflective polarization rotator consisting of silicon waveguide formed by periodically arranged rectangular air holes.
The manuscript lay-out is not well organized since experimental and result paragraphs should be added. Specifically in the experimental section the authors should add and describe all the materials and instruments used in their study; in the results and discussion section the authors should add the obtained results, discussing on them. Therefore, modify the manuscript lay-out following the reported indications.
In the introduction, the authors should add a part in which describe the function mechanism of reflective polarization rotators, adding references.
Please, briefly describe the etching process to obtain the silicon waveguide.
How was the dimension of etched rectangular air holes obtained? Please, add the characterization of the obtained device and discuss.
Please, add separately in the manuscript the equations.
Please, define the TE and the TM modes.
How was scanned the period? Discuss.
Which instrument was used to perform the reflection measurements? Add and discuss.
Which is the measurements reproducibility? Discuss.
For all quantities reports the units of measurement.
The English style should be improved.
I can accept with major revisions.
Reviewer 2 Report
The paper reports on integrated optics structure capable of rotating the plane of polarization in reflection. The structure is made of rectangular air holes in Si and pure Si rectangles. In my opinion the structure is simply a folded Solc structure with rocking angle of 45 degress of the optic axis. The authors can simply homogenize the structure and they can easily see that in one period the optic axis is along the diagonal containing the Si rectangles while the low index axis (ordinary) is along the diagonal containing air rectangles. In this sense the phenomenon observed is not new, this is simply the reflective polarization rotator of Abdulhalim reported a while ago in two different forms, see papers:
I, Abdulhalim, Reflective polarization conversion Fabry–Perot
resonator using omnidirectional mirror of periodic
anisotropic stack, Optics Communications 215 (2003) 225–230.
I. Abdulhalim, Omnidirectional reflection from anisotropic periodic dielectric
stack, Optics Communications 174 2000. 43–50.
These paper report exactly on the same phenomenon when the folded Solc structure has large birefringence and the twist angle is 45 degrees. The present paper is of interest because it is fabricating the structure in metamaterial and integrated optics form. Therefore it can be published however they should mention the homogenization to folded Solc mentioned above and the credit of the phenomenon to the original inventor.
Reviewer 3 Report
The authors propose a periodically L-shaped silicon wavegude.
The paper deals with a standard problem and is suitable for publication after the following issue.
1) how are chosen the parameters of the L-structure.
2) the optimal solution is given by periodically structure or exists other configurations, as aperiodic structure or procedures of optimization.
Round 2
Reviewer 1 Report
In my opinion, the revised version of the manuscript can be accepted to be published.
Author Response
In my opinion, the revised version of the manuscript can be accepted to be published.
Answer: Thanks for the positive comment of Reviewer.
Reviewer 2 Report
The authors did not respond to the comments properly. There is no rush, take your time and do the work properly. I suggest the following:
1. Add two arrows to show the direction of the optic axis (e-ray polarization) and the O-wave polarization to illustrate better the twisted Solc structure. Perhaps the e-axis is along the diagonal of the Si boxes while the O-axis is along the air boxes. Do that for one period to show the 45 degress twist.
2. The credit is not given properly to the author of references 27-28. They did not even mention that the 1st reflective polarization rotator was reported in these works. This must be done, otherwise it is in the limit of plagiarism. They can do it the way they wish but here is a paragraph to help them formulate it:
"In [27-28], Abdulhalim reported reflective polarization rotator (RPR) for the first time using the folded Solc structure with 45 degrees twist and high birefringence. In [27] Abdulhalim RPR is demonstrated near the photonic bandgap where omnidirectional reflection occurs, while in [28] it is demonstrated as resonant reflective peaks of Fabry-Perot structure. While Abdulhalim RPR was the 1st reported using folded Solc structure, ours is the first in planar waveguide form utilizing properties of metamaterial structures. The layers thicknesses, the high birefringence and the 45 degrees twist are the conditions postulated by Abdulhalim to observe the RPR and here our designed metamaterials satisfies these conditions."
Author Response
To Reviewer 2:
- Add two arrows to show the direction of the optic axis (e-ray polarization) and the O-wave polarization to illustrate better the twisted Solc structure. Perhaps the e-axis is along the diagonal of the Si boxes while the O-axis is along the air boxes. Do that for one period to show the 45 degree twist.
Answer: In birefringent materials, there are two O-axes and one E-axis. However, in the etched waveguide, we can only have two axes defined by the two electric field polarization directions of the propagating modes with different effective refractive indexes. We define the two axes as the slow and fast axes instead of O-axis and E-axis.
We have added a drawing of one period etched waveguide with the slow and fast axes in Fig. 1. Since the SiO2 substrate is under the silicon waveguide, the etched waveguide is not perfectly orthogonally symmetrical. The angles q for the two orthogonal eigenmodes involving in the polarization rotation is not 45 degrees. The angle q defined by the electric field polarization direction of the etched waveguide eigenmode to the vertical axis of the waveguide chip is also illustrated in Fig. 1.
- The credit is not given properly to the author of references 27-28. They did not even mention that the 1st reflective polarization rotator was reported in these works. This must be done, otherwise it is in the limit of plagiarism. They can do it the way they wish but here is a paragraph to help them formulate it:
"In [27-28], Abdulhalim reported reflective polarization rotator (RPR) for the first time using the folded Solc structure with 45 degrees twist and high birefringence. In [27] Abdulhalim RPR is demonstrated near the photonic bandgap where omnidirectional reflection occurs, while in [28] it is demonstrated as resonant reflective peaks of Fabry-Perot structure. While Abdulhalim RPR was the 1st reported using folded Solc structure, ours is the first in planar waveguide form utilizing properties of metamaterial structures. The layers thicknesses, the high birefringence and the 45 degrees twist are the conditions postulated by Abdulhalim to observe the RPR and here our designed metamaterials satisfies these conditions."
Answer: Since the SiO2 substrate is under the silicon waveguide, the etched waveguide is not perfectly orthogonally symmetrical. The angles q to the vertical axis of the waveguide chip for the two orthogonal eigenmodes involving in the polarization rotation is not 45 degrees. Per the reviewer’s comment, we added the description in Introduction in Page 2.
In literature, Abdulhalim reported the reflective polarization rotator for the first time using the folded Šolc structure with 45 degree twist and high birefringence. [25, 26] In Ref. [25], the reflective polarization rotator of Abdulhalim was demonstrated near the photonic bandgap where the omnidirectional reflection [27-30] occurs. While in Ref. [26], it was demonstrated as resonant reflective peaks of Fabry-Pérot structure. While the reflective polarization rotator of Abdulhalim was the first reported using folded Šolc structure, ours is the first in integrated optical waveguide form utilizing properties of photonic crystal structures. The layers thicknesses, the high birefringence and the 45 degree twist were the conditions postulated by Abdulhalim to observe the reflective polarization rotator. Here our photonic crystal structure followed the Abdulhalim’s design with the twist angle determined by the waveguide mode.
Round 3
Reviewer 2 Report
Following the latest modification the paper can be published now.